# Protein Intake Estimated from Brief-Type Self-Administered Diet History Questionnaire and Urinary Urea Nitrogen Level in Adolescents

**DOI:** 10.3390/nu11020319

**Published:** 2019-02-01

**Authors:** Masayuki Okuda, Keiko Asakura, Satoshi Sasaki

**Affiliations:** 1Graduate School of Sciences and Engineering for Innovation, Yamaguchi University, 1-1-1 Minami-Kogushi, Ube 755-8505, Japan; 2Department of Environmental and Occupational Health, School of Medicine, Toho University, 5-21-16 Omori-Nishi, Ota-ku, Tokyo 143-8540, Japan; jzf01334@nifty.ne.jp; 3Department of Social and Preventive Epidemiology, School of Public Health, The University of Tokyo, 7-3-1 Hongo, Bunkyo-ku, Tokyo 113-0033, Japan; stssasak@m.u-tokyo.ac.jp

**Keywords:** adolescent, biomarker, food frequency questionnaire, protein intake, urine, validity

## Abstract

Our aim was to assess the validity of the brief-type self-administered diet history questionnaire (BDHQ15y) to estimate the protein intake in 248 Japanese secondary school students (mean age = 14.2 years), using urinary biomarkers as references. Participants provided three samples of overnight urine for measurement of urea nitrogen and creatinine levels, underwent anthropometric measurements, and answered the questionnaires. Additionally, 58 students provided 24-h urine specimens. A significant correlation was observed between excretion of urea nitrogen in overnight and 24-h urine specimens (*ρ* = 0.527; *p* < 0.001), with biases ≤5.8%. The mean daily protein intake estimated from urinary biomarkers was 76.4 ± 20.4 g/d in males and 65.4 ± 16.9 g/d in females, and the mean protein intake estimated from the BDHQ15y (PRT_bdhq_) was 89.3 ± 33.7 g/d in males and 79.6 ± 24.6 g/d in females. Crude and energy-adjusted coefficients of correlation between PRT_bdhq_ and protein intake estimated from urinary biomarkers were 0.205 (*p* = 0.001; 0.247 for males and 0.124 for females), and 0.204 (*p* = 0.001; 0.302 for males and 0.109 for females), respectively. The BDHQ15y is a low-cost tool to assess protein intake of a large population, instead of a weakness of overestimation.

## 1. Introduction

Dietary intake is closely related with health outcomes in school-age children, which includes risks of cardiovascular, digestive, bone, and allergic diseases [1,2,3,4]. Dietary habits of macronutrient and food intake are known to be retained from childhood to adulthood [5,6,7], and unhealthy dietary habits are associated with cardiovascular risks in adults [8].

The school period provides adolescents with an opportunity to learn about healthy nutritional habits, which can be crucial for their health outcomes in adulthood. In order for the implementation of health education programs, a food frequency questionnaire (FFQ) is a valuable means to evaluate dietary habits among adolescents, as they are easy to administer and to answer, and they allow the nutritional assessment of a large population with a relatively low cost. On the other hand, they are susceptible to variables like cognitive performance, social preference, and body and health status [9]. Thus, the validity of FFQs should be carefully evaluated before their use in a given cultural background or age group.

Repeated 24-h diet recall and dietary records are usually used as references for FFQs, but they are vulnerable to underreporting or overreporting [10]. The brief-type self-administered diet history questionnaire (BDHQ) was validated for the Japanese adult population using diet records, and blood and urine biomarkers of dietary nutrients as references [11,12,13], and showed moderate to sufficient acceptability for epidemiological surveys [11,12]. For Japanese adolescents, the BDHQ was modified (BDHQ15y), considering the dietary habit of this population, and validated using serum concentrations of lipophilic vitamins, namely carotenoids and tocopherols, and fatty acids, as references [13]. However, its validity concerning metabolites of nutrients remains to be examined. Urinary nitrogen, a biomarker of dietary protein intake [14], is an easily available reference. This is because overnight urine is a commonly used clinical sample for annual health check-ups at school, in accordance with the School Health and Safety Law in Japan, and measurement of urea nitrogen is frequently used as a clinical test, mainly due to its low cost.

Energy and nutrient intake in adolescents is influenced by growth [15,16] and physical activity [17,18]. In adults, daily urinary excretion of sodium can be estimated using anthropometric and urinary data [19,20,21]. When it comes to adolescents, physical activity should also be considered [18]. Likewise, the dietary protein intake in adolescents is probably influenced by physical activity. The aim of this study was to assess the validity of the BDHQ15y in estimating the protein intake in Japanese secondary school students, using urinary urea nitrogen as the reference standard and considering possible confounders.

## 2. Materials and Methods

### 2.1. Subjects

The methods for recruitment and urinalysis are fully described elsewhere [18,22]. A total of 320 secondary school students from the Suo-Oshima town, Yamaguchi prefecture, Japan, were eligible for this study. Those who agreed to participate (*N* = 276) provided written informed assent and their legal guardians provided written informed consent. The primary object of this sample was to estimate sodium intake of adolescents [18]. The participants had no diagnosis of diabetes, hypertension, or kidney disease; they answered the questionnaires, underwent anthropometric measures, and provided urine samples. The study was approved by the Institutional Review Board of the Yamaguchi University Hospital (technical opinion number H25–87) and the Ethics Committee of the University of Tokyo, Faculty of Medicine (technical opinion number 10259).

### 2.2. Urinalysis

Analysis of 24-h urine is the standard method to estimate the daily protein intake, but can be difficult for many individuals. Thus, 24-h urine samples were obtained twice from 68 participants and overnight urine was obtained three times from all 276 participants for this purpose.

On the day before sampling, the participants were provided with 10-mL test tubes, 1-L plastic bottles, and a diary to record the start and end times of collection and every voiding time. They were asked to void urine into paper cups, take 7–8 mL of urine into a test tube after they awoke (overnight urine) on the first day, and collect all 24-h urine in bottles until the first void in the next morning. The second 24-h urine specimen was obtained one week later, and the subsequent overnight urine specimens were also sampled in volumes of 7–8 mL. The interval between each overnight urine collection was at least three days. Both overnight and 24-h urine specimens were transferred to the school. The 24-h urine volume was measured, and an aliquot of 7–8 mL was used for the analyses. The specimens collected in test tubes were frozen until analysis at the LSI Medience Corporation (Tokyo, Japan). The concentration of urea nitrogen (mg/dL) was determined using the urease leucine dehydrogenase method, and the concentration of creatinine (mg/dL) was determined using an enzymatic method. The daily urinary excretion of urea nitrogen (UUN_24h_) in grams per day (g/d) was measured in 24-h urine specimens and adjusted according to the following equation:
UUN_24h_ (g/d) = 24-h urinary urea nitrogen (mg/dL) × 24-h urine volume (L) × 24 h/collection period (h; finish time − start time)/100.(1)

### 2.3. Estimation of Daily Urea Nitrogen Excretion from Overnight Urine

The daily urinary excretion of urea nitrogen was measured based on overnight urine (UUN_on_) according to the following equation:
UUN_on_ (g/d) = urinary urea nitrogen (mg/dL) × estimated daily creatinine excretion (mg/d)/urinary creatinine concentration (mg/dL)/1000.(2)

The daily excretion of creatinine in 24-h urine specimens (UCr_24h_) was calculated according to the equations proposed by Mage [23], Moriyama [24], Kawasaki [19], and Tanaka [20] (Appendix B), and each equation was compared to the equation below.
UCr_24h_ (mg/d) = 24-h urinary creatinine (mg/dL) × 24-h urine volume (L) × 24 h/collection period (h; finish time − start time) × 10.(3)

### 2.4. Estimation of the Protein Intake from Measurement of Urinary Urea Nitrogen

The protein (PRT) intake was estimated by measuring the urinary urea nitrogen both in 24-h urine (PRT_24h_) and overnight urine (PRT_on_) specimens, assuming that (1) urea nitrogen constitutes 85% of nitrogen [25], (2) 81% of nitrogen intake is excreted in urine [14], and (3) nitrogen constitutes 16% of protein.
PRT = UUN/(0.85 × 0.81 × 0.16).(4)

Variance of protein intake was adjusted as usual intake (Methods, Appendix A), and protein intake was compared to the recommended dietary allowance (RDA) to satisfy the daily needs in 97.5% of people: 60 g/d for boys aged 12–14 years, and 55 g/d for girls of the same age [26].

### 2.5. Anthropometric Measurements

Anthropometric measurements were carried out at the school, with the help of the school’s nurses, within two weeks of the first urine collection. For body weight measurement, participants were asked to wear light clothes, and for body height measurement, they were asked to take their shoes off. Body mass index (kg/m^2^) was calculated as weight (kg)/height (m)^2^.

### 2.6. Questionnaire

Physical activity was assessed by asking how many minutes per week (min/wk) the participant spent doing light-, moderate-, and vigorous-intensity physical activities (LPA, MPA, and VPA, respectively) [27]. We asked how many minutes they took on their way from home to school on foot or bicycle. The total time of moderate-to-vigorous-intensity physical activity (MVPA) per week (METs-min/wk, where MET is metabolic equivalent) was calculated as 4.9 × MPA (min) + 8.9 × VPA (min) + 3.6 × walk (min) × 2 × 5 (days) + 6.2 × bike (min) × 2 × 5 (days) [27].

Age was calculated (to the first decimal place) from the difference between the first day of urine sampling and the birth date. Definition of pubertal maturation relied on the self-reported maturation of voice (for boys) and self-reported menarche (for girls). The participants were asked to inform the grade in which they were when they entered puberty and were divided into two groups: individuals who had and who had not entered puberty.

The dietary intake was assessed using the BDHQ15y, which contained four pages with 90 items addressing foods and dietary habits of the previous month. Nutrient intake was estimated using the BDHQ15y and the Standard Tables of Food Composition in Japan [28], which showed that the Spearman’s correlation coefficients between estimated dietary intake and blood biomarkers in individuals aged 13–14 years were 0.26–0.34 for carotenoids, and 0.22–0.48 for marine n-3 polyunsaturated fatty acids [13]. The BDHQ for adult had moderate correlations of nutrient intake (0.50–0.64), compared with estimation from a 16-day dietary record [12]. Dietary energy (kcal/day; 1 kcal = 4.184 kJ) and protein (PRT_bdhq_; g/day) intakes were estimated, and protein intake was also expressed as the percentage of total energy intake using an energy density method (%E).

### 2.7. Statistical Analysis

Exclusion criteria for analysis were (1) 24-h urine sampling period less than 20 h (*N* = 5), (2) 24-h urine volume lower than 600 mL (considered as incomplete collection) (*N* = 3), (3) overnight urine collected after less than 6 h of sleep (*N* = 1), and (4) energy intake (estimated from the BDHQ) <0.5 times the energy required for the lowest physical activity level, and >1.5 times the energy required for the highest physical activity level [26] (*N* = 14). Out of 276 students, 28 were excluded, and the final sample comprised 248 students (118 males and 130 females), with 58 students (21 males and 37 females) being able to provide both 24-h and overnight urine specimens, and 190 students (97 males and 93 females) being able to provide overnight urine only.

Values were expressed as means ± standard deviation (SD), unless otherwise specified. The Student’s *t*-test or the chi-square test was used to compare the results between two independent groups, and a paired *t*-test was performed when comparing of urea nitrogen and creatinine concentrations between 24-h and overnight urine samples, and PRT between BDHQ and urine estimates. Estimates were evaluated by comparing with references calculated from urine using a mean difference, Spearman’s correlation coefficients (*ρ*), an intra-class correlation coefficient (ICC (3,1)) as a validity coefficient [29], Passing–Bablok regression (Methods, Appendix A) [30], and a Bland–Altman plot [31]; comparison between estimates vs. references was estimated vs. measured creatinine, UUN_on_ vs. UUN_24h_, and protein intake was estimated from BDHQ15y vs. calculated from urine. Spearman’s correlation coefficients were used to investigate the correlation between protein intake estimated from the measurement of urinary biomarkers and possible confounders. Spearman’s correlation coefficients between protein intake estimated from BDHQ15y and urinary biomarkers were adjusted for age, MVPA, body weight, and body height. Furthermore, coefficients were corrected for attenuation due to day-to-day variability with a deattenuation factor calculated as $\sqrt{1+\delta_{w}/\delta_{b}\times 1/k}$ derived from a random-effects model, where *δw/δb* corresponds to the within/between-individual variance ratio, and *k* corresponds to the number of replications [32]. Participants were classified into five ordered subgroups according to the values of each estimate (BDHQ and urinary biomarkers), which were distributed into quintiles, and the agreement between the results of BDHQ and urinalysis was examined: “concordance” when a student was in the same or adjacent ordered subgroups of two measurements, and “discordance” when a student was in three or four away subgroups). Statistical analyses were performed using SAS version 9.4 (SAS Institute Japan Ltd., Tokyo, Japan), and a result was considered statistically significant at *p* < 0.05.

## 3. Results

The mean age of the participants was 14.2 ± 0.8 years old (males) and 14.2 ± 0.9 years old (females) (range = 12.7–15.8 in both sexes; Table 1). Male subjects were found to be more physically active than female subjects (6694.9 ± 4018.0 vs. 4347.9 ± 2820.0 METs-min/w). The subjects (*N* = 28) excluded from the analysis were similar to the subjects for analysis in terms of age, body height, body weight, and physical activity (*p* > 0.39), except for energy intake estimated from the BDHQ (3548.8 kcal; *p* < 0.001) No statistically significant difference was found in any variable between individuals who were able to provide 24-h urine and those who were not able to do so (*p* = 0.81–0.88), except for the urinary creatinine concentration (172.3 mg/dL ± 66.3 vs. 151.1 mg/dL ± 57.9, *p* = 0.029), and the rate of individuals in puberty (89% vs. 76%, *p* = 0.012), which were higher in individuals who were able to provide overnight urine only. The mean 24-h urine volumes were 952.5 mL (male subjects) and 974.2 mL (female subjects). The 24-h urine was collected during a mean period of 22.6 h by both males and females. Urea nitrogen and creatinine concentrations in 24-h urine were 941.7 mg/dL and 121.2 mg/dL (males), and 844.7 mg/dL and 110.7 mg/dL (females), which were lower than those found in overnight urine (differences, 301.7 mg/dL, and 36.6 mg/dL, respectively; both *p* < 0.001).

The mean UCr_24h_ was 1003.6 ± 228.8 mg/d (range: 493.1–1609.8; Table 2). Among the four equations used to calculate UCr_24h_, Tanaka’s equation showed the lowest bias (4.9 mg/d) and highest correlation coefficients (*ρ* = 0.601, and ICC = 0.684). Also, a Bland–Altman plot showed that 96% of the estimates using Tanaka’s equation were within the range of ±1.96 × SD (Figure 1a). Kawasaki’s equation showed the highest bias and lowest correlation coefficients. A Passing–Bablok regression line for Tanaka’s equation had a slope, 0.989, and an intercept, 9.201, which closely overlapped on an identity line (Appendix A). Thus, Tanaka’s equation was adopted to estimate the concentration of urinary creatinine and levels of protein excretion.

In students who provided 24-h urine specimens (*N* = 58), the mean UUN_24h_ was 8.7 ± 2.4 g/d (males) and 7.5 ± 2.1 g/d (females), and the mean UUN_on_ was 9.2 ± 2.6 g/d (males) and 7.6 ± 1.7 g/d (females) (Table 3). The biases of UUN_on_ were 0.1–0.5 g/d (1.7–5.8%). UUN_24h_ showed a positive correlation with UUN_on_ (*ρ* = 0.527, *p* < 0.001) and ICC (*ρ* = 0.636). These UUN results corresponded to values of protein intake ranging from 67.9 to 83.4 g/d (Table 3, Figure 1b). Considering all 248 subjects, the mean PRT_on_ estimated from UUN_on_ was 76.4 ± 20.4 g/d and 65.4 ± 16.9 g/d in males and females, respectively. After protein intake was adjusted as usual intake, 13.6–19.3% of males and 22.9–27.8% of females were found to have insufficient protein intake (Appendix A).

Spearman’s coefficients of correlation between protein intake estimated from urinary biomarkers and other variables are shown in Table 4. MVPA was significantly correlated with PRT_24h_ (*ρ* = 0.381, *p* = 0.003), but its correlation with PRT_on_ became smaller (*ρ* = 0.143, *p* = 0.024). Moderate correlations were observed between PRT_on_ and body height and weight (*ρ* = 0.463, *p* < 0.001), but significant correlations were not observed between PRT_24h_ and body height and weight (*ρ* = 0.179–0.215, *p* > 0.1). Age was positively correlated with both PRT_24h_ and PRT_on_ in males (*ρ* = 0.353, *p* = 0.116, and *ρ* = 0.220, *p* = 0.017, respectively).

The mean protein intake estimated from the BDHQ15y (PRT_bdhq_) was higher than the values estimated from PRT_24h_ and PRT_on_ (differences, 12.7 g/d, *p* = 0.005, and 13.6 g/d, *p* < 0.001, respectively). Biases were 12.9–20.5 g/d in males, and 8.3–14.2 g/d in females (Table 3), and both PRT_bdhq_ and biases were higher in males than in females. PRT_bdhq_, whether crude or energy-adjusted, was significantly correlated with PRT_on_, when considering all participants and males (Table 5; *ρ* = 0.204–0.302; *p* < 0.01). The ICC, considering all subjects, was 0.274, and a Bland–Altman plot showed that the bias increased as the mean of PRT_bhdq_ and PRT_on_ increased (Figure 1d). Although the crude coefficient of correlation between PRT_bdhq_ and PRT_24h_ (*ρ* = 0.239, ICC = 0.350) was higher than the coefficients between PRT_bdhq_ and PRT_on_ (*ρ* = 0.205, ICC = 0.274), there were no significant correlations between these two variables, even after adjustments (Figure 1c). After classification of participants into five ordered subgroups (according to PRT_bdhq_, PRT_on_, and PRT_24h_), 51.7–62.1% of participants had concordant results, while 10.3–24.1% had discordant results. PRT_bdhq_ and PRT_24h_ had the highest concordant level.

## 4. Discussion

### 4.1. Protein Intake

The subjects of this study took 64.7–78.6 g/d of urine-estimated protein, which indicated that 13.6–27.8% of them had insufficient protein intake. For children and adolescents, protein is essential for growth. However, the rate of protein deposition was not considered for the estimation of protein intake. Garlick reported that protein deposition on growth is 0.041 and 0.031 g/kg (body weight)/day in 11–15-year-old boys and girls, respectively [33]. Based on this, protein deposition was estimated to be 1.5–2.2 g/kg/day with mean body weights. These values corresponded to 2.2–2.9% of the protein intake estimated from urinary biomarkers; thus, more adolescents may have insufficient protein intake than estimated above.

### 4.2. Urinary Creatinine

Urinary creatinine was estimated using four different equations. Mage’s and Moriyama’s equations were previously developed for subjects aged 2–18 [24] and 3–18 years old [23], respectively. Mage’s equation had smaller biases when compared to Moriyama’s equation, but both had larger biases when compared to Tanaka’s equation. Tanaka’s and Kawasaki’s equations were previously developed for estimation of creatinine excretion in adults. Tanaka’s equation was used for the analysis of spot urine specimens in an epidemiological study [20], and Kawasaki’s equation was used for analysis of second morning urine specimens collected during fasting in health check-ups [19]. As urine in this study was sampled during fasting, we expected Kawasaki’s equation to be more accurate for our analyses. Urea nitrogen and creatinine concentrations in urine may vary during the day, and concentrations may be different among overnight urine and first and second morning urine specimens. In participants who provided 24-h urine specimens, the urea nitrogen/creatinine ratios were slightly different between the 24-h and overnight urine specimens on the same day (7.6–7.8 and 7.7–7.9 mg/dL, respectively).

### 4.3. Physical Activity

Our results revealed that protein intake was associated with physical activity, which is consistent with the results of a previous report addressing energy and sodium intakes [17,18]. Exercise-induced energy expenditure is not compensated for by increasing energy intake in a short time interval, such as 1–7 days [34]. Nevertheless, it is reasonable to consider that energy expenditure and intake are balanced in adolescents with normal body weight and those with stable body weight for a long period. The levels of physical activity are largely variable among secondary school students [27]; thus, students with higher energy expenditure may have a higher dietary intake and, parallelly, a higher nutrient intake.

### 4.4. BDHQ Validity

Urinary nitrogen is used as a recovery biomarker of protein intake [14]. Regarding adult individuals, moderate correlation coefficients (0.23–0.64) were previously observed between the protein intake estimated from FFQ and from urinary biomarkers [35,36,37,38,39,40,41]. Although a meta-analysis of FFQs developed for adolescents indicated fair correlation coefficients with dietary records or 24-h recalls (pooled coefficients, 0.5, and 95% confidence intervals, 0.37–0.62) [42], few reports assessed the validity of FFQs for estimation of protein intake in adolescents using urinary nitrogen as the reference standard. A British study involving adolescents aged 11–13 years old found a correlation coefficient of −0.20; however, such a coefficient was not adjusted for intra-individual variance, as only one urine sample per participant was evaluated [43].

Using diet records as the reference standard, the BDHQ version for adults was shown to underestimate both energy and protein intakes (−9.2% to −7.7% of the standard, and −6.3 to −5.5%, respectively) [12]. In a study involving adult women using urinary biomarkers as the reference standard, the protein intake was shown to be even more underestimated by the BDHQ (−15.8% to −9.5%) [44]. On the other hand, coefficients of correlation between the protein intake estimated from the BDHQ and from diet records were found to be moderate in adults (0.24–0.26), and improved after adjustment for energy (0.35–0.38) [12], although the coefficients of correlation between the estimates from the adult BDHQ and from urine biomarkers were low (0.00–0.17) [44]. In this study, however, the BDHQ15y was found to overestimate the protein intake. A meta-analysis of FFQs developed for adolescents also indicated overestimation [42]. Protein deposition accounted for 7.3–17.1% of biases between the estimates from BDHQ15y and from urinary biomarkers in this study, but could not explain the overestimation. The BDHQ15y does not inquire portion sizes, and various settings of the portion size in a food frequency questionnaire are known to cause an unspecified bias in energy intake [45,46].

Biases of PRT_bdhq_ were higher in males than in females. Furthermore, these biases were proportional ones, and were exacerbated in males with high intake (Figure 1c,d and Appendix A). Adolescents with high physical activity might intake more main staples such as rice and noodles than expected because of high energy need. The BDHQ could not adequately assess physical activity levels of adolescents and, therefore, a similar pattern of energy-generated nutrients might apply to all subjects. In these circumstances, the BDHQ may underestimate carbohydrate and overestimate protein in the subjects with large amount of intake.

Very few previous studies evaluated FFQs for adolescents separately for males and females [42]. Coefficients of correlation between PRT_bdhq_ and either PRT_24h_ or PRT_on_ were higher in males than in females. Male subjects were found to be engaged in MVPA for longer periods than female subjects, which may lead to a large variance in the estimated amount of protein intake in males. Among individuals who were excluded from the study, overestimation of energy intake was more frequent in males than in females. In the original sample (*N* = 276), the values of energy and protein intake were 2900.1 kcal/d and 97.6 g/d (males) and 2186.7 kcal/d and 79.0 g/d (females). Although the exclusion of participants with extreme values improved the correlation coefficients to a small extent, these coefficients remained smaller when compared to adult individuals. Epidemiological studies involving adolescents may require a larger sample size than studies involving adults to investigate nutritional effects on health [10,47]. Even so, the coefficients of correlation between the protein intake estimated from the BDHQ and from urinary biomarkers obtained in this study are similar to those found when serum concentrations of carotenoids were used as the reference standard (0.26–0.34) [13].

The associations of PRT_24h_ and PRT_on_ with PRT_bdhq_ showed slightly different patterns; the coefficients of correlation between energy-adjusted PRT_bhdq_ and PRT_24h_ were <0.1 in all subjects and males. This is partly explained by the difference in urea nitrogen/creatinine ratios between the 24-h and overnight urine specimens, mentioned above, but a possible reason is a matter of speculation in this sample population. On the other hand, coefficients of correlation between PRT_bdhq_ and PRT_on_ were low and insignificant in females. Most values of PRT_on_ and PRT_24h_ were positively associated with physical activity, but only PRT_on_ of females showed an inverse correlation in this population. The BDHQ15y must be carefully used in a female population at present, and warrants a future study of different female populations.

### 4.5. Strengths and Limitations of the Study

Repeated urine sampling and the high participation rate were the strengths of this study, but some limitations may have hindered the interpretation of the results. Firstly, the small sample size from a single town may not adequately represent the Japanese population. Secondly, to evaluate the BDHQ validity, 24-h urine specimens should be obtained from all subjects. Since the students collected 24-h urine specimens in a cooperative manner, we could apply 24-h urine sampling to measure other nutrients.

## 5. Conclusions

We could estimate the protein intake of secondary school students from biomarkers measured in overnight urine samples. Estimation of the protein intake from overnight urine sampled during school health check-ups can allow a larger population survey nationwide. We also evaluated the protein intake estimated from the BDHQ15y in comparison with estimates from biomarkers measured in overnight urine; it exhibited a weakness of overestimation of protein intake, but showed weak but significant correlations between the results. This means that this FFQ can assess variety of protein intake in an adolescent population. The BDHQ15y is an easy and low-cost tool for assessment of the dietary intake in adolescents and can also be used to assess the protein intake, in addition to the measurement of serum concentrations of previously validated nutrients (carotenoids, tocopherols, and fatty acids) [13].

## Figures and Tables

**Figure 1 nutrients-11-00319-f001:**
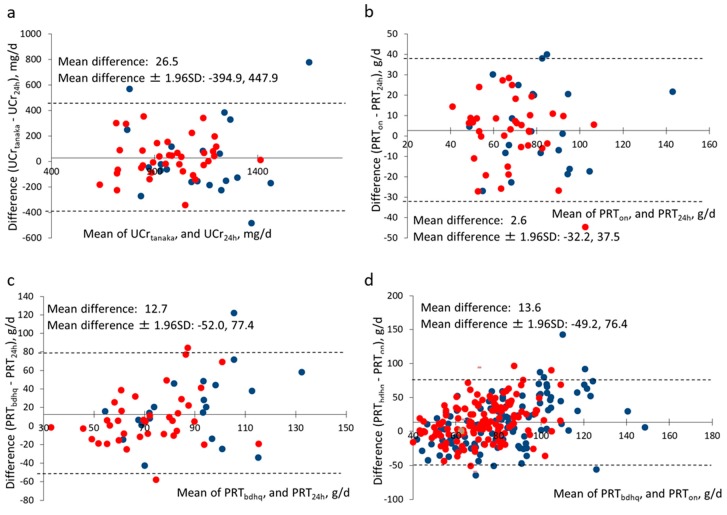
Bland–Altman plots. Estimates were compared with references; horizontal axes correspond to means of estimates and standards, and vertical axes correspond to differences [estimates − standards]. Solid horizontal axes are placed at the mean differences, and dashed lines correspond to the mean ± 1.96 × SD. Blue “●” and red “●” circles indicate male and female subjects, respectively. Estimates and references in each panel are (**a**) daily creatinine excretion estimated by Tanaka’s equation (UCr_tanaka_) and measured in 24-h urine (UCr_24h_; *N* = 58), (**b**) protein intake estimated from biomarkers measured in overnight urine (PRT_on_) and 24-h urine (PRT_24h_; *N* = 58), (**c**) protein intake estimated from the brief-type self-administered diet history questionnaire (BDHQ; PRT_bdhq_) and PRT_24h_ (*N* = 58), and (**d**) protein intake estimated from the BDHQ (PRT_bdhq_) and PRT_on_ (*N* = 248).

**Table 1 nutrients-11-00319-t001:** Characteristics of the subjects. Values are expressed as means ± standard deviation, or frequencies (percentages). MVPA—moderate-to-vigorous-intensity physical activity; METs—total time of MVPA.

	All (*N* = 248)	Males (*N* = 118)	Females (*N* = 130)	*p*-Value
Age (years)	14.2 ± 0.9	14.2 ± 0.9	14.2 ± 0.9	0.973
Body height (cm)	157.8 ± 7.8	161.4 ± 8.2	154.5 ± 5.7	0.001
Body weight (kg)	50.6 ± 10.3	52.5 ± 12.1	48.9 ± 8.0	0.006
Body mass index (kg/m^2^)	20.4 ± 3.3	20.1 ± 3.6	20.6 ± 2.9	0.312
Pubertal change				
No	35	19 (15%)	16 (11%)	
Yes	213	99 (85%)	114 (89%)	
MVPA (METs-min/wk)	5455.4 ± 3638.0	6651.1 ± 4026.9	4370.1 ± 2853.0	0.001
Overnight urine				
Urea nitrogen (mg/dL)	1207.7 ± 296.9	1260.7 ± 282.2	1159.7 ± 302.7	0.007
Creatinine (mg/dL)	167.4 ± 65.0	172.1 ± 60.0	163.1 ± 69.1	0.276
24-h urine		*N* = 21	*N* = 37	
Collection period (h)	22.6 ± 0.9	22.6 ± 1.1	22.6 ± 0.8	0.811
Urine volume (mL)	966.3 ± 374	952.5 ± 320.1	974.2 ± 403.4	0.833
Urea nitrogen (mg/dL)	879.8 ± 288.6	941.7 ± 292.2	844.7 ± 284.6	0.222
Creatinine (mg/dL)	114.5 ± 42.2	121.2 ± 39.9	110.7 ± 43.5	0.367

*p*-Values indicate comparison between genders. MVPA, moderate or vigorous physical activity; MET, metabolic equivalent.

**Table 2 nutrients-11-00319-t002:** Daily urinary creatinine excretion (*N* = 58).

	Mean ± SD	Difference	Spearman’s Correlation		Intraclass Correlation
	(mg/d)	(mg/d)	Coefficient	*p*-Value	
UCr_24h_	1009.9 ± 229.5				
Estimated UCr_24h_					
Mage’s equation [23]	951.7 ± 190.6	−58.2	0.607	<0.001	0.645
Moriyama’s equation [20]	1231.1 ± 384.0	221.2	0.560	<0.001	0.606
Kawasaki’s equation [19]	1321.8 ± 360.7	311.9	0.622	<0.001	0.669
Tanaka’s equation [24]	1036.4 ± 241.5	26.5	0.628	<0.001	0.669

UCr_24h_ (daily urinary creatinine excretion); Difference (difference of estimates from from UCr_24h_).

**Table 3 nutrients-11-00319-t003:** Estimates of daily urine nitrogen excretion and dietary protein intake (means ± SD). BDHQ—brief-type self-administered diet history questionnaire.

	All	Males	Females
Participants who provided 24-h urine	*N* = 58	*N* = 21	*N* = 37
24-h urine			
Urea nitrogen (g/d)	7.9 ± 2.3	8.7 ± 2.4	7.5 ± 2.1
Protein (g/d)	71.7 ± 20.4	78.5 ± 21.8	67.9 ± 18.8
Overnight urine			
Urea nitrogen (g/d)	8.2 ± 2.2	9.2 ± 2.6	7.6 ± 1.7
Protein (g/d)	74.4 ± 20.0	83.4 ± 23.6	69.3 ± 15.9
BDHQ			
Protein (g/d)	84.4 ± 30.1	99.0 ± 32.5	76.2 ± 25.5
Energy (kcal/d)	2410.7 ± 757.7	2899.3 ± 750.9	2133.4 ± 614.3
Adjusted protein (%E)	14.0 ± 2.3%	13.6 ± 2.5%	14.2 ± 2.2%
All participants	*N* = 248	*N* = 118	*N* = 130
Overnight urine			
Urea nitrogen (g/d)	7.8 ± 2.1	8.4 ± 2.3	7.2 ± 1.9
Protein (g/d)	70.6 ± 19.4	76.4 ± 20.4	65.4 ± 16.9
BDHQ			
Protein (g/d)	84.2 ± 29.6	89.3 ± 33.7	79.6 ± 24.6
Energy (kcal/d)	2407.6 ± 751.0	2663.2 ± 785.6	2175.7 ± 637.4
Adjusted protein (%E)	14.0 ± 2.5%	13.3 ± 2.5%	14.7 ± 2.4%

**Table 4 nutrients-11-00319-t004:** Factors associated with protein intake estimated from urinary biomarkers.

	PRT_24h_			PRT_on_		
	All *N* = 58	Males (*N* = 21)	Females (*N* = 37)	All *N* = 248	Males (*N* = 118)	Females (*N* = 130)
g/day		78.5 ± 21.8	67.9 ± 18.8		76.4 ± 20.4	65.4 ± 16.9
Age	0.027	0.353	−0.113	0.015	0.220	−0.139
	(0.840)	(0.116)	(0.507)	(0.811)	(0.017)	(0.115)
MVPA	0.381	0.343	0.299	0.143	0.141	−0.014
	(0.003)	(0.128)	(0.072)	(0.024)	(0.127)	(0.878)
BMI	0.097	0.192	0.037	0.285	0.372	0.305
	(0.469)	(0.404)	(0.829)	(<0.001)	(<0.001)	(<0.001)
Body weight	0.179	0.233	0.062	0.463	0.449	0.431
	(0.178)	(0.310)	(0.716)	(<0.001)	(<0.001)	(<0.001)
Body height	0.215	−0.011	0.231	0.463	0.339	0.431
	(0.105)	(0.962)	(0.168)	(<0.001)	(<0.001)	(<0.001)
Pubertal change						
Yes	72.3 ± 19.4	78.7 ± 30.2	68.8 ± 10.7	72.6 ± 27.2	73.1 ± 33.2	71.9 ± 18.6
No	71.5 ± 20.9	78.5 ± 19.7	67.6 ± 20.9	70.3 ± 17.9	77.1 ± 17.1	64.4 ± 16.5
	(0.929)	(0.683)	(0.468)	(0.739)	(0.075)	(0.463)

Values are expressed as Spearman’s correlation coefficients or means (compared using *t*-tests). The numbers between parentheses correspond to the *p*-values. PRT_24h_ (protein intake estimated from biomarkers measured in 24-h urine), PRT_on_ (protein intake estimated from biomarkers measured in overnight urine), MVPA (amount of moderate- or vigorous-intensity physical activity, given in METs-min/w), BMI (body mass index, given in kg/m^2^).

**Table 5 nutrients-11-00319-t005:** Spearman’s coefficients of correlation between values of protein intake estimated from urinary biomarkers and protein intake estimated from BDHQ.

	Protein Estimated from Urinary Biomarkers (g/d)		
	All	Males	Females
Protein from BDHQ	PRT_24h_ (subjects who provided 24-h urine)		
	*N* = 58	*N* = 21	*N* = 37
Crude (g/d)	0.239	0.325	0.247
	(0.075)	(0.164)	(0.148)
Energy density (%E)	−0.028	−0.097	0.189
	(0.833)	(0.682)	(0.229)
	PRT_on_ (all subjects)		
	*N* = 248	*N* = 118	*N* = 130
Crude (g/d)	0.205	0.247	0.124
	(0.001)	(0.008)	(0.161)
Energy density (%E)	0.204	0.302	0.109
	(0.001)	(0.001)	(0.218)

PRT_24h_, PRT_on_, and PRT_bdhq_ (protein intake estimated from biomarkers measured in 24-h urine samples, biomarkers measured in overnight urine, and BDHQ15y, respectively). Spearman’s correlation coefficients were adjusted for age, MVPA, body weight, body height, and sex, if necessary, and corrected for attenuation.

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
