# Peer review of "Protein Intake Estimated from Brief-Type Self-Administered Diet History Questionnaire and Urinary Urea Nitrogen Level in Adolescents"

_nutrients, 2019, doi:10.3390/nu11020319_

Reviewer 1 Report

The study by Okuda et al compares the self-administered diet history 15 questionnaire (BDHQ15y) to assess protein intake with protein intake assessed by urinary nitrogen excretion. The study is well described, and the results are presented clearly.
The aim of the study was to assess the validity of the BDHQ15y questionnaire. While carefully performed, the true validation is only possible in a subset of the samples, since only 58 students had 24h urine collections (the golden standard).
Comment 1:
Figure 1:  The x and y-axis titles are difficult to read. I would recommend increasing title sizes and the text size in the graphs.
Comment 2:
Line 44: not just underreporting. Other relevant limitations to FFQ are, overreporting, changes in diet due to self-reflections, errors in portion size estimates, and socially desirable answers
Comment 3:
Line 207: (ρ = 463) à (ρ = 0.463)
Comment 4:
Line 101 “Estimation of the protein intake from measurement of urinary urea nitrogen”, in this section the assumptions for calculating protein intake are shown, but the used formula itself not. Is there a reason that the formula itself is omitted?
Comment 5:
I cannot seem to find the correlation coefficient + p-value between protein intake estimated from 24h and overnight urine.
It would be nice to see further comparison of overnight urine and 24h urine estimated protein, perhaps in the form of a Passing-Bablok graph.
Comment 6:
In comparing protein intake based on 24h urine and BDHQ15y there is a large bias of 12.7 g/24h. This difference cannot fully be attributed to protein deposition, so the BDHQ15y appears to overestimate protein intake in Japanese students. This seems like one of the major results in terms of assessing the validity of the questionnaire, but this conclusion is, however, not drawn in the conclusion section of the article, nor the abstract. Why not?
Comment 7:
In figure 1d, there appears to be a significant proportionality bias. The higher the average intake, the more protein intake from BDHQ15y appears to overestimate protein intake from overnight urines, especially in males. This is correctly noted in line 219 of the results section. Can you explain this finding?
As this proportionality bias (mainly in males) reflects the validity of the BDHQ15y questionnaire, I believe it is worth noting in the conclusion section of the article.
Linear regression analysis between difference (PRTbdhq – PRTon) and mean (PRTbdhq and PRTon) could be used to quantify the proportionality bias.

Comment 8: 
In females BDHQ15y estimated protein intake did not associate with overnight urine estimated protein intake, nor 24h urine estimated protein intake. To the reader, it seems, based on this data, that the BDHQ15y is not reliable for estimating protein intake in females. What is your opinion on this?

Author Response

We appreciate your review of our article. The reviewers’ comments are helpful to improve our manuscript. We have revised the manuscript with red fonts, and responded each reviewer’s comments as follows;

Comment 1:
Figure 1:  The x and y-axis titles are difficult to read. I would recommend increasing title sizes and the text size in the graphs.

We have enlarged the font size in Figure 1.

Comment 2:
Line 44: not just underreporting. Other relevant limitations to FFQ are, overreporting, changes in diet due to self-reflections, errors in portion size estimates, and socially desirable answers

We have revised this sentence.

Repeated 24-h diet recall and dietary records are usually used as references for FFQs, but they are vulnerable to underreporting or overreporting.” Line 44.

Comment 3:
Line 207: (ρ = 463) à (ρ = 0.463)

              We have corrected it to 0.463. Line 227.

Comment 4:
Line 101 “Estimation of the protein intake from measurement of urinary urea nitrogen”, in this section the assumptions for calculating protein intake are shown, but the used formula itself not. Is there a reason that the formula itself is omitted?

We have added the formula.

PRT = UUN / ( 0.85 × 0.81 × 0.16). Line 109.

Comment 5:
I cannot seem to find the correlation coefficient + p-value between protein intake estimated from 24h and overnight urine.
It would be nice to see further comparison of overnight urine and 24h urine estimated protein, perhaps in the form of a Passing-Bablok graph.

Since protein intake was a simple multiple of urea nitrogen, correlation coefficients and p-values were same as those of urea nitrogen. Passing-Bablok regression has been calculated and added it in the Methods, Results, and Supplementary materials.

Methods, “Estimates were evaluated in comparing with references calculated from urine using a mean difference, Spearman’s correlation coefficients (ρ), an intra-class correlation coefficient (ICC (3,1)) as a validity coefficient [29], Passing Bablok regression (Supplementary materials) [30], and a Bland-Altman plot [31]” Lines 152–155.

A Passing Bablok regression line for the Tanaka’s equation had a slope, 0.989, and an intercept, 9.201, which closely overlapped on an identity line (Supplementary Results, Table S2, and Figure S1).” Lines 191-193.

Comment 6:
In comparing protein intake based on 24h urine and BDHQ15y there is a large bias of 12.7 g/24h. This difference cannot fully be attributed to protein deposition, so the BDHQ15y appears to overestimate protein intake in Japanese students. This seems like one of the major results in terms of assessing the validity of the questionnaire, but this conclusion is, however, not drawn in the conclusion section of the article, nor the abstract. Why not?

We have added senteces about bias

Abstract, “The BDHQ15y, is a low-cost tool to assess protein intake of a large population, instead of a weakness of overestimation.” Line 27.

Conclusion, “; it exhibited a weakness of overestimation of protein intake, but showed weak but significant correlations between the results. It means that this FFQ can assess variety of protein intake in an adolescent population.” Lines 357–359.

Comment 7:
In figure 1d, there appears to be a significant proportionality bias. The higher the average intake, the more protein intake from BDHQ15y appears to overestimate protein intake from overnight urines, especially in males. This is correctly noted in line 219 of the results section. Can you explain this finding?
As this proportionality bias (mainly in males) reflects the validity of the BDHQ15y questionnaire, I believe it is worth noting in the conclusion section of the article.

Linear regression analysis between difference (PRTbdhq – PRTon) and mean (PRTbdhq and PRTon) could be used to quantify the proportionality bias.

We have added a paragraph in the Discussion.

              Biases of PRTbdhq were higher in males than in females. Furthermore, these biases were proportional ones, and exacerbated in males with high intake (Figure 1c, 1d S1c, and S1d). Adolescents with high physical activity might intake more amount of main staples such as rice and noodles than expected because of high energy need. The BDHQ could not adequately assess physical activity levels of adolescents, and therefore a similar pattern of energy-generated nutrients might apply to all subjects. In these circumstances, the BDHQ may underestimate carbohydrate and overestimate protein in the subjects with large amount of intake.” Lines 315–321.

Comment 8: 
In females BDHQ15y estimated protein intake did not associate with overnight urine estimated protein intake, nor 24h urine estimated protein intake. To the reader, it seems, based on this data, that the BDHQ15y is not reliable for estimating protein intake in females. What is your opinion on this?

We added sentences in the Discussion.

 “On the other hand, coefficients of correlation between PRTbdhq and PRTon were low and insignificant in females. Most of PRTon and PRT24h were positively associated with physical activity, but only PRTon of females showed inverse correlation in this population. The BDHQ15y must be carefully used in a female population at present, and warrants a future study of different female populations.” Lines 340-344.

Reviewer 2 Report

The authors estimated protein intake from a Brief Diet History Questionnaire (BDHQ) and from urinary biomarkers, in order to validate the BDHQ as a tool to asses protein intake in adolescents. The BDHQ15y overestimated protein intake compared with urinary biomarkers, and the correlation coefficient between the two sources was low-moderate (0.2 overall, 0.3 and 0.1 in males and females, respectively) although statistically significant.

Aim

The authors should also state the secondary aim as disclosed in the discussion, i.e. estimating protein intake from urine biomarkers and asses deficiency status in healthy Japanese adolescents.

Methods

The authors should provide information whether they calculated the required sample size.

The authors should provide references for the methods used to determine the concentration of urea nitrogen, and creatinine and for the formula to calculate UUN24h, UUNon and UCr24h (g/d)

Estimation of the protein intake from measurement of urinary urea nitrogen: the authors refer to reference 14 and 25, although in discussion they state: “The equation used to estimate the protein intake from UUN in this study was adopted from a study with adults [20]”. Please check and modify accordingly.

In the questionnaire paragraph please report which source of data (nutrient database) was used to link food items and calculate nutrient intake.

Statistical analysis: report characteristics of excluded students in the results section.

Report in methods also how the comparison between protein intake from urinary biomarkers was compared with RDA, and cite the software used (PC side version 1.02, Iowa State University, Statistical Laboratory, Iowa, USA).

Results

When interpreting results (e.g. “Urea nitrogen and creatinine concentrations in 24-h urine were lower than those found in overnight urine”; “The mean protein intake estimated from the BDHQ15y (PRTbdhq) was higher than the values estimated from PRT24h and PRTon”) please provide also the p value, either in text or in tables.

In the text the authors state “Male subjects were found to be more physically active than female subjects (6694.9 ± 4018.0 vs. 4347.9 ± 2820.0 METs-min/w)”, however values reported in table 1 are slightly different and no p values are reported.

Similarly, in the text we read “The mean UCr24d was 1003.6 ± 228.8 mg/d (range: 493.1–1609.8; Table 2).”, but in table 2 different values are presented. The same applies for the correlation coefficient and ICC of the Tanaka’s equation in the text and table 2. Please check carefully and make sure that figures are consistent between text and tables.

Reports in the results section also the data on the proportion of individuals meeting or not the RDA and the related table 6, which is currently in the discussion section.

Discussion

BDHQ Validity: The correlation coefficient for the validation of the BDHQ for adolescents may be discussed more in details, together with the pooled results of the cited meta-analysis by Tabacchi et al 

Tables

Table 1. Add an extra initial column with results for all participants and a final column with p values for differences among males and females.

Table 3. Add an extra initial column with results for all participants. The authors may explore the possibility to combine table 3 and 5.

Author Response

We appreciate your review of our article. The reviewers’ comments are helpful to improve our manuscript. We have revised the manuscript with red fonts, and responded each reviewer’s comments as follows;

Aim

1. The authors should also state the secondary aim as disclosed in the discussion, i.e. estimating protein intake from urine biomarkers and asses deficiency status in healthy Japanese adolescents.

The aim of this section may confuse the reader, so we have added Supplementary materials to describe this section (the methods, and results in the previous version Discussion). We have simply described the methods, and the results in the text (simply to validate protein intake measurement using urine, but not to elucidate deficiency), and revised the Discussion.

Methods, “Variance of Protein intake was adjusted as usual intake (Supplementary Methods), and protein intake was compared to the recommended dietary allowance (RDA) to satisfy the daily needs in 97.5% of people: 60 g/d for boys aged 12–14 years, and 55 g/d for girls of the same ages [26].” Lines 110–112.

Result, “After protein intake was adjusted as usual intake, 13.6–19.3% of males, and 22.9–27.8% of females were found to have insufficient protein intake (Supplementary Results, Table S1).” Lines 209–211.

Discussion, “The subjects of this study took 64.7–78.6 g/d of urine-estimated protein, which indicated 13.6–27.8% of them had insufficient protein intake.” Lines 248–249.

Methods

2. The authors should provide information whether they calculated the required sample size.

We have added the sentence.

The primary object of this sample was to estimate sodium intake of adolescents [18].” Lines 68–69.

3. The authors should provide references for the methods used to determine the concentration of urea nitrogen, and creatinine and for the formula to calculate UUN24h, UUNon and UCr24h (g/d)

These formulae are unit conversions.

4. Estimation of the protein intake from measurement of urinary urea nitrogen: the authors refer to reference 14 and 25, although in discussion they state: “The equation used to estimate the protein intake from UUN in this study was adopted from a study with adults [20]”. Please check and modify accordingly.

This sentence has been deleted to focus on the discussion about the BDHQ.

5. In the questionnaire paragraph please report which source of data (nutrient database) was used to link food items and calculate nutrient intake.

We have added a reference.

Nutrient intake was estimated using the BDHQ15y and the Standard Tables of Food Composition in Japan [28],” Line 132.

6. Statistical analysis: report characteristics of excluded students in the results section.

We have added the sentence about exclusion.

The subjects excluded from the analysis were similar to the subjects for analysis in terms of age, body height, body weight, and physical activity (P > 0.39), except for energy intake estimated from the BDHQ (3548.8 kcal; P < 0.001).” Lines 174–177.

7. Report in methods also how the comparison between protein intake from urinary biomarkers was compared with RDA, and cite the software used (PC side version 1.02, Iowa State University, Statistical Laboratory, Iowa, USA).

 This section may confuse the reader, so we have simply described the methods, and added Supplementary materials to describe this section.

Methods, “Variance of Protein intake was adjusted as usual intake (Supplemetary materials), and protein intake was compared to the recommended dietary allowance (RDA) to satisfy the daily needs in 97.5% of people: 60 g/d for boys aged 12–14 years, and 55 g/d for girls of the same ages [27].” Lines 110–112.

Results

8. When interpreting results (e.g. “Urea nitrogen and creatinine concentrations in 24-h urine were lower than those found in overnight urine”; “The mean protein intake estimated from the BDHQ15y (PRTbdhq) was higher than the values estimated from PRT24h and PRTon”) please provide also the p value, either in text or in tables.

P values have been added.

Methods, “a paired t-test was performed when comparing of urea nitrogen and creatinine concentrations between 24h and overnight urine samples, and PRT between BDHQ and urine estimates.“ Lines 150-152.

Results, “Urea nitrogen and creatinine concentrations in 24-h urine were lower than those found in overnight urine (differences, 301.7 mg/dL, and 36.6 mg/dL, respectively; both P < 0.001).” Lines 183–186

The mean protein intake estimated from the BDHQ15y (PRTbdhq) was higher than the values estimated from PRT24h and PRTon (differences, 12.7 g/d, P = 0.005, and 13.6 g/d, P < 0.001, respectively).” Lines 231–233.

9. In the text the authors state “Male subjects were found to be more physically active than female subjects (6694.9 ± 4018.0 vs. 4347.9 ± 2820.0 METs-min/w)”, however values reported in table 1 are slightly different and no p values are reported.

Similarly, in the text we read “The mean UCr24d was 1003.6 ± 228.8 mg/d (range: 493.1–1609.8; Table 2).”, but in table 2 different values are presented. The same applies for the correlation coefficient and ICC of the Tanaka’s equation in the text and table 2. Please check carefully and make sure that figures are consistent between text and tables.

Reports in the results section also the data on the proportion of individuals meeting or not the RDA and the related table 6, which is currently in the discussion section.

Discussion

10. BDHQ Validity: The correlation coefficient for the validation of the BDHQ for adolescents may be discussed more in details, together with the pooled results of the cited meta-analysis by Tabacchi et al 

              We have added discussion about correlation coefficients.

Although a meta-analysis of FFQs developed for adolescents indicated fair correlation coefficients with dietary records or 24h recalls (pooled coefficients, 0.5, and 95% confidence intervals, 0.37–0.62) [42]” Lines 294-296.

Very few previous studies evaluated FFQs for adolescents separately for males and females [42].” Lines 322-323.

On the other hand, coefficients of correlation between PRTbdhq and PRTon were low and insignificant in females. Most of PRTon and PRT24h were positively associated with physical activity, but only PRTon of females showed inverse correlation in this population. The BDHQ15y must be carefully used in a female population at present, and warrants a future study of different female populations.” Lines 340–344.

Biases of PRTbdhq were higher in males than in females. Furthermore, these biases were proportional ones, and exacerbated in males with high intake (Figure 1c, 1d S1c, and S1d). Adolescents with high physical activity might intake more amount of main staples such as rice and noodles than expected because of high energy need. The BDHQ could not adequately assess physical activity levels of adolescents, and therefore a similar pattern of energy-generated nutrients might apply to all subjects. In these circumstances, the BDHQ may underestimate carbohydrate and overestimate protein in the subjects with large amount of intake.” Lines 315–321.

Tables

11. Table 1. Add an extra initial column with results for all participants and a final column with p values for differences among males and females.

We have revised Table 1 as per se.

12. Table 3. Add an extra initial column with results for all participants. The authors may explore the possibility to combine table 3 and 5

We have added a column for all participants. Although we have tried a combined table, which seemed complicated because of having different units, and involvement of confounders, we have chosen separated tables.

Reviewer 3 Report

Dear Editor,

I am pleased to review an assigned manuscript. Overall, a proposed article looks good to me – Authors really did a wonderful job and presented very nice & relevant literature. The English is generally satisfactory, although there are some places where corrections and/or changes are required. The study does have few minor issues. The authors should improve the paper following these suggestions.

Abstract need bit of attention and should cover theme of whole manuscript

The introduction should be better organized. Some of the sentences are not well structured, should be clarified and rewritten. It is advice to link the story in a better way in an introduction to convey a proper message to readers

Result and discussion is good, it’s well written and well organized but you can focus only latest reference in discussion if you want

Please recheck the reference style, some of the references are not according to the journal instructions

In general, manuscript looks good to me and authors did a good job. I am happy to review the revised version.

Author Response

We appreciate your review of our article. The reviewers’ comments are helpful to improve our manuscript. We have revised the manuscript with red fonts, and responded each reviewer’s comments as follows;

1. Abstract need bit of attention and should cover theme of whole manuscript

We have added a weakness of the BDHQ in the Abstract, and transferred the section of comparison with RDA into the Supplementary Materials to focus on the BDHQ.

2. The introduction should be better organized. Some of the sentences are not well structured, should be clarified and rewritten. It is advice to link the story in a better way in an introduction to convey a proper message to readers

We have revised the Introduction.

3. Result and discussion is good, it’s well written and well organized but you can focus only latest reference in discussion if you want

We have revised the first paragraph of the Discussion, deleted Table 6.

4. Please recheck the reference style, some of the references are not according to the journal instructions

We have corrected the reference style.